# Gene–Gene Interactions Reduce Aminoglycoside Susceptibility of *Pseudomonas aeruginosa* through Efflux Pump-Dependent and -Independent Mechanisms

**DOI:** 10.3390/antibiotics12010152

**Published:** 2023-01-11

**Authors:** Aswin Thacharodi, Iain L. Lamont

**Affiliations:** Department of Biochemistry, University of Otago, Dunedin 9054, New Zealand

**Keywords:** *Pseudomonas aeruginosa*, efflux pump, MexZ repressor, FusA1, AmgS, gene expression, antibiotic resistance, aminoglycoside resistance, tobramycin, amikacin, gentamicin

## Abstract

*Pseudomonas aeruginosa* causes a wide range of acute and chronic infections. Aminoglycosides are a cornerstone of treatment, but isolates are often resistant. The purpose of this research was to better understand the genetic basis of aminoglycoside resistance in *P. aeruginosa*. Bioinformatic approaches identified mutations in resistance-associated genes in the clinical isolates of *P. aeruginosa*. The common mutations were then engineered into the genome of *P. aeruginosa* reference strain PAO1. Mutations in the elongation factor gene *fusA1* caused the biggest reduction in aminoglycoside susceptibility, with mutations in the two-component regulator gene *amgS* and the efflux pump regulator gene *mexZ* having less impact. This susceptibility was further reduced by combinations of mutations. Mutations in *fusA1*, *amgS* and *mexZ* all increased the expression of the *mexXY* efflux pump that is strongly associated with aminoglycoside resistance. Furthermore, the *fusA1 amgS mexZ* triple mutant had the highest efflux pump gene expression. Engineering *fusA1* and *amgS* mutants lacking this efflux pump showed that *fusA1* and *amgS* also reduce aminoglycoside susceptibility through additional mechanisms. The *fusA1* and *amgS* mutations reduced bacterial growth, showing that these mutations have a fitness cost. Our findings demonstrate the complex interplay between mutations, efflux pump expression and other mechanisms for reducing the susceptibility of *P. aeruginosa* to aminoglycosides.

## 1. Introduction

*Pseudomonas aeruginosa* is a high-priority pathogen that causes severe opportunistic and nosocomial infections in humans [1,2]. Aminoglycosides, including tobramycin, amikacin and gentamicin, are crucial tools in antipseudomonal chemotherapy. They are used to treat a range of infections, such as nosocomial infections and pulmonary infections, in individuals with cystic fibrosis or chronic obstructive pulmonary disease [3,4,5,6,7]. However, a significant proportion of *P. aeruginosa* are resistant to multiple antimicrobials, including aminoglycosides, making the management of *P. aeruginosa* infections more challenging [8,9]. In a minority of resistant isolates, aminoglycoside resistance is due to acquired aminoglycoside modifying enzymes [10]. In the absence of these, resistance to aminoglycosides is often related to increased production of the MexXY-OprM efflux pump, which is a key contributor to intrinsic aminoglycoside resistance in *P. aeruginosa* [11,12]. This tripartite pump is composed of three protein components: MexX (periplasmic membrane fusion protein), MexY (inner membrane antiporter) and OprM (outer membrane channel) [12]. The *mexXY* genes are in an operon with expression controlled by a repressor protein MexZ, whereas the *oprM* gene is expressed from a separate operon [13,14]. Expression of the *mexXY* operon is induced by ribosome-targeting agents, such as aminoglycosides [15,16]. Mutations in *mexZ* that result in increased *mexXY* expression are considered to be a frequent cause of aminoglycoside resistance in the clinical isolates of *P. aeruginosa* [17,18,19]. Genetic variability within the MexXY efflux system may also influence substrate specificity. For example, variations at the putative substrate binding pocket of MexY have been linked to enhanced aminoglycoside resistance, indicating the role of this protein in aminoglycoside recognition and export [20].

The regulation of *mexXY* expression is complex. It can be influenced by mutations in other genes as well as *mexZ*, notably *fusA1* (encoding elongation factor G; EF-G) [21] and the *amgRS* genes that encode a two-component regulatory system [22,23]. In isolates from cystic fibrosis patients, *fusA1* is the most frequently mutated gene linked to aminoglycoside resistance [21,24]. The EF-G (FusA1) protein is involved in ribosomal elongation and recycling processes [25]. However, little is known about the phenotypic repercussions of *fusA1* mutations on the overall physiology of *P. aeruginosa* [21]. AmgRS governs an adaptive response to membrane damage that is induced by aminoglycoside-generated aberrant polypeptides [26]. Acquired aminoglycoside resistance due to mutations in the *amgS* gene arises in both experimentally evolved and clinical isolates of *P. aeruginosa* [22]. A variant of *amgS*, V121G, increased *mexXY* expression by 3-fold and enhanced aminoglycoside tolerance [22]. However, to date, the precise mechanism through which *amgS* enhances *mexXY* expression to contribute to aminoglycoside resistance is unknown. In addition, disruption of a*mgRS* genes has also been associated with enhanced resistance to tobramycin [26] and a variety of other aminoglycosides [27,28].

Although the MexXY-OprM efflux pump is a key player in the resistance of *P. aeruginosa* to aminoglycosides, the susceptibility of clinical isolates does not correlate with *mexXY* expression [29,30,31]. The lack of correlation between *mexXY* expression and aminoglycoside resistance is most likely due to the influence of other intrinsic resistance mechanisms, masking the effects of differences in MexXY-OprM efflux pump activity. However, to the best of our knowledge, interactions between different resistance mechanisms have not been investigated. The overall aim of this research was to investigate the interplay between *mexY*, *mexZ*, *amgS* and *fusA1* mutations in determining the levels of aminoglycoside susceptibility.

## 2. Results

### 2.1. Identifying Frequent FusA1 and AmgS Sequence Variants in Clinical and Environmental Isolates of P. aeruginosa 

Isolates of *P. aeruginosa* from the general environment are less likely to be resistant to antibiotics than isolates from clinical settings [32]. The analysis of MexZ and MexY in over 700 clinical and environmental isolates using PROVEAN, which identifies sequence variants likely to affect protein function, showed that significant variants in these proteins are much more frequent in clinical isolates [11]. Here, we extended this analysis to AmgS and FusA1. Significant variants in each of these proteins were present in more than 10% of the clinical isolates. The AmgR, ParR and ParS proteins, which have also been associated with aminoglycoside resistance [22,28,33], had a lower (less than 10%) frequency of predicted function-altering variants and were not further analyzed. The prevalence of AmgS and FusA1 variants in the clinical and environmental isolates is shown in Table 1. FusA1 (elongation factor G) variants were present in 28% of the clinical isolates. Most (25%) of the function-altering variants in the clinical isolates were in the domains of the protein that are involved in docking to the A-site of the ribosome [21,34] (Appendix A). Function-altering variants in AmgS were present in 10% of the clinical isolates (Table 1) (Appendix A). In contrast, less than 5% of the isolates from the general environment had predicted function-altering variants in FusA1 or AmgS, indicating that genetic alterations in these genes are uncommon in nature and more common in the clinical setting where the bacteria are more likely to be exposed to antibiotics.

### 2.2. Effects of fusA1, amgS and mexY Mutations on Aminoglycoside Resistance 

In order to quantify the effects of *amgS* and *fusA1* mutations, the most frequent sequence variants in clinical isolates (*fusA1*_R680C_ and *amgS*_V121G_, Table 1) were engineered into the genome of *P. aeruginosa* reference strain PAO1. The *amgS*_V121G_ mutant caused a 2-fold increase in the MIC for all three of the tested aminoglycosides, and the *fusA1*_R680C_ mutant resulted in a 4-fold increase in the MIC for all the aminoglycosides (Table 2). These findings are consistent with previous reports for these variants [22,24] and indicate that mutations in *amgS* and *fusA1* contribute to a reduced susceptibility to aminoglycosides. 

The most frequent sequence variant in *mexY* in the clinical isolates is a G287S variant [11] and we also engineered this mutation into the PAO1 genome. It did not affect the MIC for any of the tested antibiotics. Deletions of *mexZ* are common in clinical isolates and a deletion of *mexZ* in the PAO1 strain caused a 2-fold increase in the MIC for all the tested aminoglycosides (Table 2) [11]. 

### 2.3. Effects of Combinations of Mutations 

Pairwise combinations of *amgS*_V121G_, *fusA1*_R680C_, *mexY*_G287S_ and ∆*mexZ* mutations were engineered in PAO1 in order to determine whether combinations further reduced aminoglycoside susceptibility. An additive effect on the MICs was considered to occur when the combinations of mutations gave higher MICs than either of the individual mutations.

In most cases, double mutants had higher MICs for at least one of the tested antibiotics than either of the relevant single mutants (Table 2). For example, *fusA1_R680C_ ∆mexZ*, *fusA1*_R680C_
*mexY*_G287S_ and *fusA1*_R680C_
*amgS*_V121G_ double mutants all had higher MICs than the PAO1 *fusA1* single mutant. The exceptions to this included the ∆*mexZ mexY*_G287S_ and ∆*mexZ amgS*_V121G_ mutants, which had the same MICs as the PAO1 ∆*mexZ* single mutant.

To further investigate the *fusA1*–*mexZ* interaction, *mexZ* was also deleted from PAO1 containing the *fusA1* variants D327S or Y683D, which arose during the selection of tobramycin-resistant mutants [36]. In each case, a 2-fold increase in the MIC was observed for all the tested aminoglycosides following deletion of *mexZ*, except for tobramycin in the *fusA1_R680C_* variant (Figure 1).

Combinations of multiple mutations were engineered in order to see if they further increased the aminoglycoside MICs. A quadruple *fusA1_R680C_ amgS_V121G_ mexY_G287S_* ∆*mexZ* mutant and triple mutants containing the *fusA1_R680C_* mutation had the highest MICs. In all these cases, the MIC was higher than any of the double mutants for at least one of the tested antibiotics. In contrast, the MICs of ∆*mexZ amgS_V121G_ mexY_G287S_* were no higher than those of an *amgS_V121G_ mexY_G287S_* double mutant. Overall, *fusA1_R680C_* contributed most to increases in the MIC, with the other mutations having smaller, though significant, impacts. Notably, all the engineered triple and quadruple mutants that contained the *fusA1_R680C_* mutation reached the MIC threshold for being classified as resistant to gentamicin and amikacin.

### 2.4. The Effects of Mutations on mexXY Gene Expression

Increased tolerance to aminoglycosides can result from increased expression of the MexXY-OprM efflux pump due to mutations affecting the MexZ repressor [11]. The engineered mutants were used to investigate the effects of other mutations on the expression of the genes encoding this efflux pump.

As expected, the ∆*mexZ* mutation increased the expression of *mexXY* (Table 3). The *amgS_V121G_* mutation also elevated *mexXY* expression by 2.5-fold, similar to a 3-fold increase in the expression reported previously for this variant [23]. The *fusA1_R680C_* mutation raised *mexXY* expression by 2-fold, which was consistent with a previous report that a different *fusA1* mutation (*fusA1_P443L_*) increases *mexXY* expression [21]. These results indicate that the increased aminoglycoside MICs of *mexZ, amgS* and *fusA1* mutants (Table 2) are due, at least in part, to the increased expression of *mexXY*.

The expression of *mexXY* was then measured in mutants containing combinations of *fusA1_R680C_, amgS_V121G_* and ∆*mexZ* mutations. All the double mutants had much higher *mexXY* expression than the single mutants. For example, the ∆*mexZ fusA1_R680C_* double mutants had 17-fold higher expression than the *fusA1_R680C_* mutant, and 7-fold higher expression than the ∆*mexZ* mutant. The ∆*mexZ amgS_V121G_* double mutant also had 5-fold higher expression than the ∆*mexZ* single mutant. The triple mutant ∆*mexZ fusA1_R680C_ amgS_V121G_* had the highest expression of *mexXY.* Collectively, these data show that while mutations in *mexZ* are important contributors to increased *mexXY* gene expression, *fusA1* and *amgS* mutations also play a role.

The expression of MexXY efflux pumps does not correlate with aminoglycoside resistance in the clinical isolates of *P. aeruginosa*. This is most likely due to the multifactorial nature of resistance [11,31]. The mutants developed here are isogenic with the PAO1 strain, except for the introduced mutations. Therefore, the correlation of *mexXY* expression with the aminoglycoside MICs was examined for the engineered mutants (Appendix A). There was a clear trend towards correlation between *mexXY* expression and the MIC for all three antibiotics (tobramycin (R^2^ = 0.52), gentamicin (R^2^ = 0.67) and amikacin (R^2^ = 0.62)), but with some outlying points. 

### 2.5. The Effects of fusA1 and amgS Mutations in the Absence of mexXY Genes

The partial but incomplete correlation between the MICs and the level of *mexXY* gene expression suggested that, as well as causing an increased expression of *mexXY*, mutations in *fusA1* and *amgS* may affect the MIC by other mechanisms. To test this possibility, the *mexXY* genes were deleted in *amgS_V121G_* and *fusA1_R680C_* mutants and in the *fusA1_R680C_ amgS_V121G_* double mutant, and the effects on the MICs were determined. The deletion of *mexXY* reduced the MIC of the *fusA1* and *amgS* mutants, but the MICs were higher than for the PAO1 *mexXY* mutant (Figure 2). The deletion of *mexXY* had less of an effect in the *fusA1_R680C_ amgS_V121G_* double mutant, with the MICs for the *fusA1_R680C_ amgS_V121G_* ∆*mexXY* mutant being 32-fold higher than those of the PAO1 ∆*mexXY* mutant. These findings indicate that the effects of *fusA1* and *amgS* mutations on MICs are due, in part, to their effects on *mexXY* expression. Furthermore, especially when in combination, they increase the MIC through a *mexXY*-independent mechanism.

### 2.6. The Effects of Mutations on Growth Rates 

The ability of resistant mutants to grow in the absence of antibiotics is a critical factor in determining how effectively resistant bacterial populations will be maintained in the absence of antibiotic exposure [37]. Resistance-associated mutations can cause alterations in the proteins involved in critical biological activities, which may impact bacterial growth in the absence of antibiotics [37]. Therefore, the growth of the engineered mutants in antibiotic-free media was measured (Figure 3).

The mutations in *amgS* or *fusA1* significantly reduced growth compared to wildtype PAO1. The growth of the *fusA1_R680C_ amgS_V121G_* double mutant was further reduced and was less than either of the single mutants. The deletion of *mexXY* or *mexZ* or the presence of the *mexY_G287S_* sequence variant did not affect growth either alone or in combination with fusA1 and amgS mutations. 

## 3. Discussion 

Antibiotic resistance in *P. aeruginosa* is multifactorial [38,39], with sequence variations in numerous genes influencing the amount of antibiotics that the bacteria can tolerate. The MexXY-OprM efflux pump plays an important role in aminoglycoside resistance, but mutations in *amgS* and *fusA1* also reduce susceptibility. Here, we quantify the effects of mutations in different genes on susceptibility and investigate the interplay between different resistance-associated genes. Our findings show that mutations in *fusA1* can play a major role in reducing aminoglycoside susceptibility, but the presence of additional mutations in the *amgS, mexZ* and *mexY* genes can further increase the amount of antibiotic that the bacteria can tolerate. In addition, we show that the effects of *fusA1* and *amgS* mutations in reducing antibiotic susceptibility are due in part, but not solely, because they cause increased expression of the *mexXY* efflux pump genes. 

A mutation in the *amgS* sensor kinase gene (*amgS_V121G_*) resulted in increased *mexXY* expression, which is associated with aminoglycoside resistance (Table 2) [22,23]. However, not all the *amgS* mutants affect *mexXY* expression [23]. For example, *amgS_R182C_*, the second most common clinical variation in our clinical dataset (Table 1), does not increase *mexXY* expression. This is consistent with our finding that the *amgS_V121G_* mutation increases the MIC through both *mexXY*-dependent and -independent pathways. Deleting *mexZ* in the *amgS_V121G_* mutant increased *mexXY* expression, but did not increase the aminoglycoside MIC, demonstrating the complex relationship between *amgS* mutations, *mexXY* expression and resistance. 

Mutations in *fusA1* that are likely to affect protein function were present in 28% of the clinical isolates examined. The engineered PAO1 *fusA1_R680C_* mutants had enhanced *mexXY* expression and aminoglycoside MICs. However, the *fusA1* mutation also reduced susceptibility independent of MexXY. This was shown by the finding that the deletion of *mexXY* from a *fusA1_R680C_* mutant reduced the MIC, but not to the level of a *mexXY* mutant with wild-type *fusA1*. Spontaneously developed *fusA1* mutants can also have increased *mexXY* expression [21], but not all fusA1 variants enhance *mexXY* expression [24], further emphasizing that *fusA1* mutations can increase the MIC in a *mexXY*-independent manner. Overall, our data show that both *fusA1* and *amgS* mutations can reduce aminoglycoside susceptibility through at least two different mechanisms, one of which is the increased expression of *mexXY*. This finding explains the incomplete correlation between *mexXY* expression and MIC in the engineered PAO1 mutants.

How *fusA1* and *amgS* mutations reduce susceptibility, independent of *mexXY*, is not clear. The *fusA1_R680C_ amgS_V121G_* double mutant had a higher MIC than either of the single mutants, suggesting that they act through different mechanisms. An *amgS_V121G_* mutation reduces the membrane depolarization caused by antibiotics and this may increase aminoglycoside tolerance [23]. The proteomic and transcriptomic analysis of a *fusA1* variant (P443L) revealed increased ribosome synthesis in *P. aeruginosa* [21], which may influence the susceptibility to aminoglycosides. Alternatively, aminoglycosides inhibit protein synthesis by binding to the A-site in the ribosome [40] and mutations affecting an elongation factor protein may interfere with ribosome–aminoglycoside interactions. The proteomic and transcriptomic analysis of mutants generated in this study will help to identify how fusA1 and amgS mutations, individually and in combination, reduce aminoglycoside susceptibility independently of the MexXY-OprM efflux pump.

Chromosomal resistance mutations frequently have a detrimental effect on the host bacterium [37,41,42] and these negative effects are frequently connected with a reduction in bacterial growth rate. The a*mgS_V121G_* and *fusA1_R680C_* mutations reduced cell growth rate (Figure 3), which is consistent with the EF-G variant lowering protein synthesis and growth rates [43]. Despite this, *fusA1_R680C_* and other likely function altering *fusA1* variants were present in the genomes of 28%, and *amgS* variants in 10%, of the clinical isolates. This frequency suggests that the benefits of these sequence variants outweigh the negative effects of reduced growth rate.

In the mutants studied here, there was a partial correlation between *mexXY* expression and aminoglycoside susceptibility, but this is not the case for the clinical isolates of *P. aeruginosa* [18,29,30,31]. Our data suggest that gene–gene interactions influence the resistance phenotype directly, as well as by affecting the levels of *mexXY* expression. It seems likely that additional proteins that contribute to intrinsic resistance mask the effects of *mexXY* expression levels on aminoglycoside tolerance in clinical isolates and further research will be required to identify these. The presence of aminoglycoside-modifying enzymes [10] would also reduce the correlation between the MICs and *mexXY* expression in the clinical isolates of *P. aeruginosa*. 

## 4. Materials and Methods

### 4.1. Growth Conditions and Bacterial Strains Used in the Study

The bacterial strains and plasmids used in the study are described in Appendix A. Bacterial cells were cultured at 37 °C in Luria broth (L-broth) at 200 rpm and Luria agar (L-agar), with antibiotics as required. A tetracycline antibiotic was used to maintain or select the plasmid pEX18Tc and its derivatives in *E. coli* (12.5 µg/mL) and *P. aeruginosa* (24 µg/mL). For conjugation, an ST18 donor strain was grown with shaking (200 rpm) in L-broth supplemented with δ-aminolaevulinic acid (ALA) (50 µg/mL) and the recipient *P. aeruginosa* was grown in static conditions at 42 °C in L-broth supplemented with 0.4% KNO_3._

### 4.2. DNA Methods 

*P. aeruginosa* genomic DNA (gDNA) was extracted using the UltraClean Microbial Kit (Qiagen, Hilden, Germany) according to the manufacturer’s instructions. Plasmid DNA was extracted from JM83 *E. coli* strains using the Roche High Pure plasmid extraction kit (Basel, Switzerland). PCR-amplified products were purified using the PCR clean-up and gel extraction kit (Macherey Nagel, Dueren, Germany). The QIAquick gel extraction kit (Qiagen) was used to extract DNA from the agarose gels according to the manufacturer’s instructions.

### 4.3. Allelic Replacement and Genetic Mutants 

Using a two-step allelic replacement approach, the mutations were engineered in the *P. aeruginosa* reference strain PAO1 [44]. In brief, gDNA was extracted from the mutants carrying the mutation of interest. PCR amplification, using high-fidelity Q5 polymerase (New England Biolabs, Ipswich, MA, USA) in conjunction with appropriate primers (Appendix A), was performed to amplify ~1100 bp of DNA upstream and downstream of the target mutation, forming a ~2.2 Kb PCR product. The amplified fragments were cloned into the plasmid pEX18Tc using appropriate enzymes purchased from New England Biolabs. The resulting recombinant plasmids were then subjected to Sanger sequencing using M13 universal primers in order to confirm the presence of the desired PCR-amplified products in the recombinant plasmids and the absence of any unintended mutations. Mutation-specific screening primers (Appendix A) were used in sequencing in order to further confirm the targeted mutation. The mutations were introduced into the PAO1 genome by homologous recombination, with *E. coli* ST18 serving as the donor strain, as previously described [45]. In order to screen the mutant candidates, PCR amplification and sequencing was performed using primers specific to the mutation-containing DNA region (Appendix A). The resulting sequences were compared with the wild-type PAO1 genome using the blastX tool of NCBI in order to identify the presence of the expected variant and to confirm that no other variants had been introduced. The same approach was used to create the mutants with combinations of mutations.

### 4.4. Gene Expressional Analysis 

RT-qPCR analysis was performed as described previously [31]. Briefly, RNA was isolated from *P. aeruginosa* cultured on Mueller–Hinton (MH) agar plates grown for 16 h at 37 °C using the Qiagen RNeasy Mini kit (Qiagen). The reference genes *clpX* and *oprL* [46] and the *mexX* target gene primers are included in Appendix A. The genomic DNA dilutions of the reference strain PAO1 were used for quantifying primer efficiencies and ranged between 1.8 and 2 for all the primer pairs. Using qScript XLT cDNA SuperMix (Quantabio, Beverley, MA, USA), aliquots of RNA were reverse transcribed in order to generate cDNA, and the transcripts were measured using RT-qPCR with a LightCycler 480 SYBR Green I Master kit (Roche, Basel, Switzerland) [31]. In order to validate the amplified products, a melt curve analysis was performed and the desired PCR products were confirmed by agarose gel electrophoresis. The crossing points, target/reference ratios and melting temperatures were calculated using LightCycler 480 software. The results of the RT-qPCR analysis were derived from three biological replicate cultures, with two technical replicates for each. 

### 4.5. Whole Genome Sequences and Variant Calling 

Whole Genome sequences of 619 clinical and 172 environmental isolates of *P. aeruginosa* that are genetically diverse and well characterized [31,36,47,48] were used in this study (Appendix A). The clinical isolates of *P. aeruginosa* were acquired from individuals suffering from cystic fibrosis, chronic obstructive pulmonary disease and other serious diseases from various countries. These genome datasets were used to predict non-synonymous variants that are likely to affect protein function, as described previously [36]. In brief, sequence variants were identified and compared against a protein sequence database and the effects of the variants on sequence similarity were determined using PROVEAN [49,50]. Variants with a score of −2.5 or less were considered to be likely to alter protein function [49]. 

### 4.6. Measuring Minimum Inhibitory Concentration (MIC)

The MICs were determined for tobramycin (Mylan New Zealand Ltd., Auckland, New Zealand), gentamicin (Pfizer New Zealand Ltd., Auckland, New Zealand) and amikacin (Merck Sharp & Dohme, Auckland, New Zealand). Overnight cultures of *P. aeruginosa* grown in L-broth were diluted to 1.5 × 10^6^ CFU/mL and 5 μL aliquots were spot-plated onto Mueller–Hinton Agar (Becton Dickinson, Auckland, New Zealand) plates supplemented with doubling concentrations of antibiotic. The plates were incubated in aerobic conditions for 24 h at 37 °C. The MIC for each isolate was established as the lowest concentration that hindered observable growth while excluding solitary colonies or faint haze. The CLSI guidelines were utilized in order to classify resistant and sensitive phenotypes [50]. A tobramycin and gentamicin MIC ≥ 8 μg/mL, and an amikacin MIC ≥ 32 μg/mL, were considered clinically resistant.

### 4.7. Bacterial Growth Kinetics

The growth of the engineered mutants was analyzed as described previously [36]. In brief, overnight cultures grown in L-broth were diluted to 1.5 × 10^6^ CFU/mL and 200 μL was dispensed into transparent polystyrene 96-well plates (Corning, Durham, NC, USA). The plates were then incubated at 37 °C at 200 rpm for 24 h in a BMG FLUOstar Omega microplate reader. In order to monitor growth, an optical density (OD) at 600 nm was recorded every 30 min for 24 h. The measured optical density was then blank corrected prior to calculating the area under the curve (AUC). The AUC analysis was performed using GraphPad prism V.9. The logistic area under the curve (AUC) was utilized as a metric of growth. This metric provides a measure of growth that includes a lag phase, log phase growth rate and final cell density. The growth analysis was performed with three biological replicates (three technical replicates each).

### 4.8. Statistical Analysis and Homology Model

Graphpad Prism 9.3.0 (Graphpad Software Inc, San Diego, CA, USA) was used to run a Student’s *t*-test on the AUC data for the mutants and wildtypes. All the tests were two-tailed and *p* values less than 0.05 were deemed statistically significant. In order to predict the structure of AmgS, a homology model was made using Phyre2 [51].

## 5. Conclusions

Aminoglycoside resistance in *P. aeruginosa* is a major clinical problem. Our findings quantify the effects of combinations of mutations in *mexZ, fusA1* and *amgS* in reducing susceptibility to aminoglycosides, which was information that was previously lacking. They also show that mutations in any of these genes increase the expression of MexXY efflux pump genes. Both the MIC and efflux gene expression are highest in mutants containing combinations of *mexZ*, *fusA1* and *amgS* genes, indicating that the mutations act through different mechanisms to increase gene expression and the MIC. However, as well as affecting susceptibility through altering expression of the *mexXY* genes, *fusA1* and *amgS* mutations can reduce susceptibility through MexXY-independent mechanisms which have not yet been defined. In the long term, complete understanding of the genetic basis of resistance will enable the effectiveness of aminoglycosides to be predicted from the genome sequences of infecting *P. aeruginosa*. Predicting the effects of mutation combinations on resistance will increase the capacity to foresee and limit the emergence and spread of multidrug-resistant bacteria, as well as identifying the treatment combinations that are most effective in lowering resistance evolution.

## Figures and Tables

**Figure 1 antibiotics-12-00152-f001:**
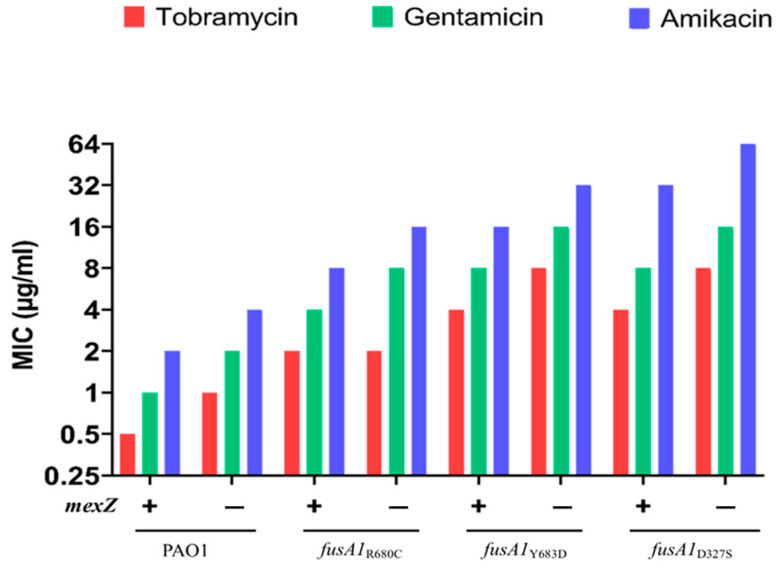
Additive effects of *fusA1* and *mexZ* mutations. The MICs of tobramycin, gentamicin and amikacin were measured for the PAO1 strain, genetically engineered f*usA1_R680C_*, tobramycin-evolved spontaneous *fusA1* variants (Y683D and D327S) and their *mexZ* isogenic mutants. The presence and absence of *mexZ* are represented as “+” and “−”, respectively.

**Figure 2 antibiotics-12-00152-f002:**
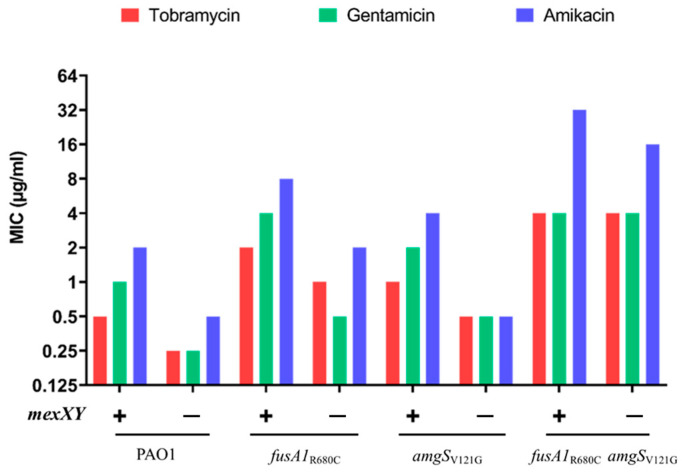
The effects of *fusA1* and *amgS* mutations in the absence of *mexXY*. The MICs of tobramycin, gentamicin and amikacin were measured for the PAO1 strain with *fusA1* or *amgS* mutations, and their isogenic *mexXY* mutants. The presence and absence of *mexXY* are represented as “+” and “−”, respectively.

**Figure 3 antibiotics-12-00152-f003:**
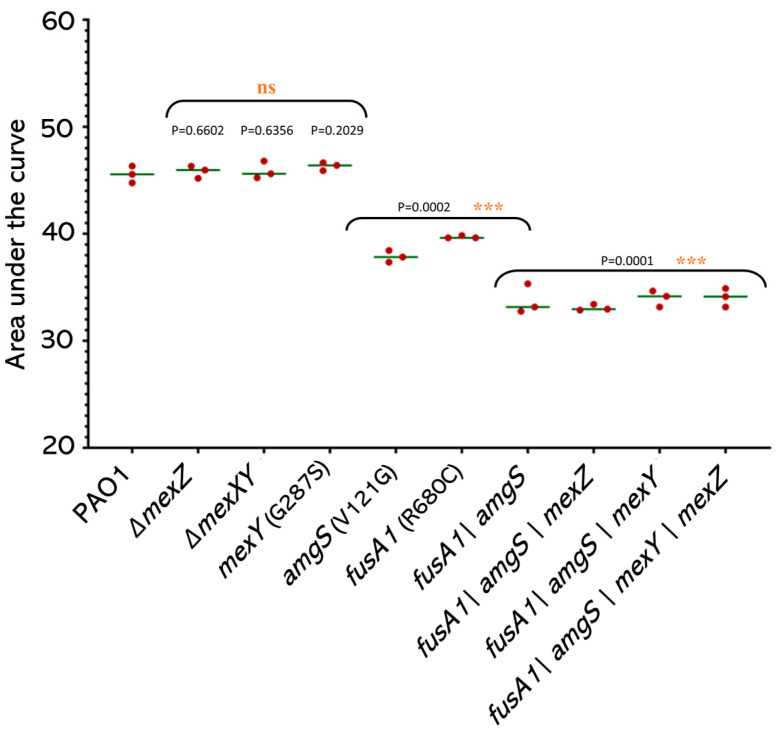
Growth of engineered mutants. Growth of three biological replicates of the PAO1 strain and isogenic mutants in L-broth, summarized as area under the curve. “ns”, no significant difference; ***, significantly different from the PAO1 strain.

**Table 1 antibiotics-12-00152-t001:** Frequency of sequence variants in clinical and environmental isolates of *P. aeruginosa*.

	Gene	C ^a^	E ^a^	Function	Frequent Variants ^b^	PROVEAN Score ^c^	Frequent Variants	PROVEAN Score
					Clinical		Environmental	
1	*fusA1*	28	1	EF-G: translation elongation factor G	R680C (14) A481V (14) M648V (13) Y552C (10)	−7.502 −3.364 −3.35 −7.533	E245I (2) C266L (2)	−3.869 −8.554
2	*amgS*	10	5	Part of two-component regulator system	V121G (26) R389S (3) R182C (3) D292G (3) P124Q (3)	−6.471 −5.431 −4.833 −3.386 −6.891	R389H (2)	−4.513

^a^ Abbreviations: C, percentage in clinical isolates; E, percentage in environmental isolates. ^b^ Most frequent PROVEAN-significant variants with their prevalence in clinical (n = 619) and environmental (n = 172) isolates shown in brackets. ^c^ Variants with a score of −2.5 or less are considered likely to affect protein function.

**Table 2 antibiotics-12-00152-t002:** MICs of engineered mutants.

Strains	Aminoglycoside MICs ^a^ (µg/mL)		
	Tob	Gen	Amik
Wild Type (WT)			
PAO1	0.5	1	2
Single mutants			
∆*mexZ*	1	2	4
*mexY* _G287S_	0.5	1	2
*fusA1* _R680C_	2	4	8
*amgS* _V121G_	1	2	4
Double mutants			
∆*mexZ mexY*_G287S_	1	2	4
∆*mexZ fusA1*_R680C_	2	8	16
∆*mexZ amgS*_V121G_	1	2	4
*amgS* _V121G_ *mexY* _G287S_	1	2	8
*mexY* _G287S_ *fusA1* _R680C_	2	4	16
*fusA1* _R680C_ *amgS* _V121G_	4	4	32
Triple mutants			
∆*mexZ fusA1*_R680C_ *amgS*_V121G_	4	8	32
∆*mexZ fusA1*_R680C_ *mexY*_G287S_	4	8	32
∆*mexZ amgS*_V121G_ *mexY*_G287S_	1	2	8
*fusA1* _R680C_ *amgS* _V121G_ *mexY* _G287S_	4	8	64
Quadruple mutant			
*fusA1*_R680C_*amgS*_V121G_*mexY*_G287S_ ∆*mexZ*	4	16	64

^a^ MIC calculated based on the median of three technical replicates. Tobramycin and gentamicin MICs ≥ 8, and amikacin MIC ≥ 32, were considered clinically resistant [35]. Abbreviations: Tob, tobramycin; Gen, gentamicin; Amik, Amikacin.

**Table 3 antibiotics-12-00152-t003:** The effect of mutations on mexX expression.

Strain	*mexX* Expression	Fold Change ^a^	Fold Changes in MIC ^a^		
			Tob	Gen	Ami
Reference strain					
PAO1	0.0007	n/a	n/a	n/a	n/a
Single mutants					
*fusA1* _R680C_	0.00142	2	4	4	4
*amgS* _V121G_	0.00173	2.5	2	2	2
∆*mexZ*	0.00379	5	2	2	2
Double mutants					
∆*mexZ fusA1*_R680C_	0.02497	35.5	4	8	8
∆*mexZ amgS*_V121G_	0.01743	25	2	2	2
*fusA1*_R680C_ *amgS*_V121G_	0.01894	27	8	4	16
Triple mutant					
∆*mexZ fusA1*_R680C_ *amgS*_V121G_	0.03383	48	8	8	16

^a^ Fold change compared to the wildtype PAO1 reference strain. MIC changes are derived from data in Table 2.

## Data Availability

The genome sequences of bacteria used in this study have been deposited with the National Centre for Biotechnology Information (NCBI). The accession numbers are listed in Appendix A.

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
