# Peer review of "Gene–Gene Interactions Reduce Aminoglycoside Susceptibility of Pseudomonas aeruginosa through Efflux Pump-Dependent and -Independent Mechanisms"

_antibiotics, 2023, doi:10.3390/antibiotics12010152_

Round 1

Reviewer 1 Report

The manuscript ‘Gene-gene interactions reduce aminoglycoside susceptibility of Pseudomonas aeruginosa through efflux pump-dependent and independent mechanisms’ is well-written and interesting research on the interplay between gene mutants involved in the resistance mechanism against aminoglycosides in Pseudomonas aeruginosa.

Consider including how the authors identified the most frequent mutations in the abstract

Since OprM is also a critical gene involved in the resistance against aminoglycosides. Whether OprM was considered for the identification of variants in the isolates?

How do the authors predict the function-altering variants? Consider including a few lines on how is the PROVEAN score calculated.

According to the authors, the fusA1 and amgS variants are already reported. Are there any reports on the combination mutants related to the resistance mechanism? Discuss the novelty of this study in the introduction

Why is mexX expression not studied in the quadruple mutant?

Since the mexXY-independent mechanism is quite interesting. Can the authors discuss or hypothesize the possible mexXY-independent mechanism?

Consider discussing the future directions of this research and the research gaps that still need to be addressed.

Minor issues:

P. aeruginosa is not italicized in some places

The manuscript has some spacing issues and minor grammar errors

Line 364 To calculate monitor growth?

Table 1 add superscript for E

Line 213 Why few strains are bold

Author Response

Response to Reviewer 1

The manuscript ‘Gene-gene interactions reduce aminoglycoside susceptibility of Pseudomonas aeruginosa through efflux pump-dependent and independent mechanisms’ is well-written and interesting research on the interplay between gene mutants involved in the resistance mechanism against aminoglycosides in Pseudomonas aeruginosa.

Thank you for your careful reading of our manuscript, and thoughtful comments and suggestions.

Consider including how the authors identified the most frequent mutations in the abstract

Thank you for this suggestion. We have adapted the abstract to include this information, while remaining within the word limit.

Since OprM is also a critical gene involved in the resistance against aminoglycosides. Whether OprM was considered for the identification of variants in the isolates?

We analysed OprM for sequence variants in the same genome set in a previous paper (Thacharodi and Lamont, J Med Microbiology 2022; cited in this manuscript [reference 11]). We did not find any significant sequence variants in OprM in the clinical isolates of P. aeruginosa.

How do the authors predict the function-altering variants? Consider including a few lines on how the PROVEAN score is calculated.

Thank you for this suggestion. We have expanded the methods section to outline the basis of the Provean method.

According to the authors, the fusA1 and amgS variants are already reported. Are there any reports on the combination mutants related to the resistance mechanism? Discuss the novelty of this study in the introduction.

Thank you for this suggestion. We have now included a sentence in the last paragraph of the introduction addressing this point.

Why is mexX expression not studied in the quadruple mutant?

mexY encodes a structural component of the MexXY-OprM efflux pump and there are no reports of mutations in mexY (or mexX or oprM) affecting efflux pump gene expression. In addition the mexY mutation had the smallest effect of any of the mutations on the MIC. mexY mutants were therefore excluded from the gene expression analysis. While there is no reason to expect that the presence of a mexY mutation would affect gene expression, we agree with the reviewer that it could be of interest to confirm this in a future study.

Since the mexXY-independent mechanism is quite interesting. Can the authors discuss or hypothesize the possible mexXY-independent mechanism?

Possible mechanisms are outlined in the discussion (lines 253-261).

Consider discussing the future directions of this research and the research gaps that still need to be addressed.

We have now expanded the discussion (lines 262-264 and 278-279) indicating the key knowledge gaps that remain to be filled.

Minor issues:

  1. aeruginosa is not italicized in some places
  2. aeruginosa is now italicized throughout the manuscript.

The manuscript has some spacing issues and minor grammar errors

We have revisited the manuscript and corrected the issues and errors that we identified.

Line 364 To calculate monitor growth?

We have corrected this typographical error

Table 1 add superscript for E

We have corrected this typographical error

Line 213 Why few strains are bold

This was a formatting error, that we have now corrected.

Reviewer 2 Report

The abstract should state briefly the purpose of the research and its benefits the scientific society.

In Introduction, the major defect of this study is the debate or argument is not clearly stated. In addition, the introduction should be clearly stated to the research questions and targets first. Then answer several questions: Why is the topic important (or why do you study it)? What are the research questions? What has been studied? What are your contributions? Why is it to propose this particular method?

Conclusions: Please provide a separate section for the conclusion. Please make sure your conclusions section underscores the scientific value-added of your paper, and/or the applicability of your findings/results. Highlights the novelty of your study.

----------------

Author Response

Response to Reviewer 2

Thank you to the reviewer for their suggestions on how to improve our manuscript.

The abstract should state briefly the purpose of the research and its benefits the scientific society.

We have adjusted the abstract to include the purpose of the research, while remaining within the 200-word limit. The last sentence of the abstract summarizes the advances in knowledge arising from our research.

In Introduction, the major defect of this study is the debate or argument is not clearly stated. In addition, the introduction should be clearly stated to the research questions and targets first. Then answer several questions: Why is the topic important (or why do you study it)? What are the research questions? What has been studied? What are your contributions? Why is it to propose this particular method?

The first part of the introduction, outlining the high significance of P. aeruginosa as a pathogen, the key role of aminoglycosides in treatment, and the clinical problem of aminoglycoside resistance outlines the importance of the topic and why we studied it. We have now expanded the last paragraph of the Introduction to clearly state the gap in current knowledge, as suggested by Reviewer 1. The research question (Aim) is summarized in the last sentence of the introduction. The contributions from the research are summarized in the Abstract, as well as being described in the results and the discussion. The methods that we used were the most direct way to investigate the incidence and effects of individual and combinations of mutations of resistance and gene expression. It is not clear what other methods could be used and so we have not discussed other methods.

Conclusions: Please provide a separate section for the conclusion. Please make sure your conclusions section underscores the scientific value-added of your paper, and/or the applicability of your findings/results. Highlights the novelty of your study.

We have added a Conclusions section, along the lines requested by the reviewer.

Reviewer 3 Report

This paper describes the research on the most frequent mutations in resistance-associated genes in clinical isolates of P. aeruginosa. The study included engineering of the mutations into the genome of P. aeruginosa reference strain and their association with aminoglycoside resistance was evaluated. The authors provide interesting and important findings (i.e. besides mutations in mexZ, mutations in fusA1 and amgS contribute to increased mexXY gene expression).

The article is concise, well written and has adequate introduction in the topic, presentation of material and methods and the results. Discussion is appropriate.

Therefore, I recommend it for publishing in Your Journal.

Author Response

Response to Reviewer 3

This paper describes the research on the most frequent mutations in resistance-associated genes in clinical isolates of P. aeruginosa. The study included engineering of the mutations into the genome of P. aeruginosa reference strain and their association with aminoglycoside resistance was evaluated. The authors provide interesting and important findings (i.e. besides mutations in mexZ, mutations in fusA1 and amgS contribute to increased mexXY gene expression).

The article is concise, well written and has adequate introduction in the topic, presentation of material and methods and the results. Discussion is appropriate.

Therefore, I recommend it for publishing in Your Journal.

We thank the reviewer for reading our manuscript and for their positive comments. There are no comments that need to be addressed.